# A Transmission Efficiency Evaluation Method of Adaptive Coding Modulation for Ka-Band Data-Transmission of LEO EO Satellites

**DOI:** 10.3390/s22145423

**Published:** 2022-07-20

**Authors:** Zhongguo Wang, Fan Lu, Dabao Wang, Xiao Zhang, Jionghui Li, Jindong Li

**Affiliations:** 1Institute of Remote Sensing Satellite, China Academy of Space Technology, Beijing 100094, China; wang_zhongguo@sohu.com (Z.W.); cast_wdb@126.com (D.W.); appspring2022@163.com (X.Z.); ljdcast@163.com (J.L.); 2Beijing Institute of Spacecraft System Engineering, Beijing 100094, China; lijionghui@126.com; 3Beijing Engineering Research Center of EMC & Antenna Test, Beijing 100094, China

**Keywords:** adaptive coding modulation, transmission efficiency, Ka-band data transmission, EO satellites

## Abstract

Nowadays low Earth orbit (LEO) Earth observation (EO) satellites commonly use constant coding modulation (CCM) or variable coding modulation (VCM) schemes for data transmission to ground stations (G/S). Compared with CCM and VCM, the adaptive coding modulation (ACM) could further improve the data throughput of the link by making full use of link resource and the time-varying characteristics of atmospheric attenuation. In order to comprehensively study the data transmission performance, one new index which could be utilized as a quantitative index for the satellite-to-ground data transmission scheme selection, the transmission efficiency factor (TEF) of LEO satellites is proposed and defined as “the product of the link availability and the average useful data rate”. Then, the transmission efficiency of CCM, VCM and ACM at typical G/S with different weather characteristics at Ka-band is compared and analyzed. The results show that ACM is more suitable for the G/S with moderate and abundant rainfall. Compared with the CCM of MCS 28, for Beijing G/S and Sanya G/S, ACM not only improves the transmission efficiency with the TEF increased by 3.62% and 24.51%, respectively, but also improves the link availability with the outage period reduced by 82.47% and 75.18%, respectively.

## 1. Introduction

For LEO EO satellites, the orbit characteristics lead to limited time for satellite-to-ground data transmission [1]. Technological advancements in satellite engineering have made it feasible to develop EO satellite systems with high spatial, spectral, temporal and radiometric resolutions [2,3,4,5]. With the development of multifunction and high-resolution payloads, there is an obvious boost of the on-board storage and downlink transmission data volume. Therefore, it is necessary to increase the satellite-to-ground data transmission rate and meet the increasing demand for the realization of high-resolution earth observation missions [6].

For the mostly used X-band [7] CCM data transmission scheme, as applied with LEO EO satellites WorldView-3 [8] and GaoFenDuoMo-1 [9], the link condition variance during a single transmission arc segment can reach 10 dB with the elevation angles ranging between 5° and 90° [10]. Additionally, the VCM data transmission scheme is proposed for better use of this margin caused by dynamic link conditions. The GaoFen-7 satellite of China, launched in 2019, adopts the X-band VCM for the first time [11,12].

However, the 375 MHz bandwidth of X-band has been saturated for higher data rate requirements. Therefore, the Ka-band with 1.5 GHz bandwidth [5] has become the development trend of data transmission [13,14]. The Ka-band data-transmission suffers a much greater impact of atmospheric environment, whose rain attenuation could exceed 10 dB [15]. This unpredictable variance of the link condition leads to even larger waste of link resources compared with traditional CCM or VCM scheme. Additionally, more link margins need to be designed and reserved to overcome the larger atmospheric attenuation at Ka-band. Based on the above consideration, a close-loop ACM system for Ka-band data transmission could be realized by adding a feedback channel with real-time signal-to-noise ratio (SNR) estimation. According to the real-time feedback of the dynamic downlink condition caused by distance and atmospheric attenuation, the appropriate modulation and coding mode can be adaptively selected at the transmitter, so as to improve the data throughput of the downlink [16]. ACM has been proposed in some communication protocols by European Telecommunications Standards Institute (ETSI) and Consultative Committee for Space Data Systems (CCSDS) [17,18,19].

Some literature has developed research on Ka-band CCM, VCM and ACM systems of LEO satellites. Ref. [16] compares the data throughput of CCM, VCM and ACM at Sanya G/S under sunny and rainstorm conditions with one transmission arc segment, and only partially indicates the effect of various attenuations. Refs. [20,21] analyze the characteristics and link condition of the VCM system, while Ref. [22] focuses on comparing and analyzing different estimation algorithms of the received SNR of the ACM system. Based on the modulation and coding schemes (MCS) recommended by CCSDS using 26.65 GHz signals of Sentinel-1, Ref. [23] analyzes the link availability of CCM, S-VCM, D-VCM and ACM at Matera G/S and Svalbard G/S in Europe. The whole transmission arc segment is divided into six sectors of equal duration, and transmission efficiency analysis is carried out based on the minimum link availability of 99.5% regarding to the specific selected G/S, which might not be applicable to all scenarios.

However, the correlation between transmission efficiency and link availability has not been noticed in the above papers. For LEO EO satellites, if only pursuing high link availability, sufficient margin needs to be reserved to compensate the atmospheric attenuation, especially for small elevation angles, which will result in lower useful data rate and could not meet the requirement of high-rate data transmission. On the contrary, if only pursuing high useful data rate, little margin could be reserved to compensate the atmospheric attenuation, which will result in smaller link availability, especially for small elevation angles, and the outage period might be so long and seriously affect the user’s data receiving task. Therefore, there should be a balance between the useful data rate and the link availability to maximize the transmission efficiency of LEO EO satellites, and this is the basis of the TEF proposed in this paper.

In this paper, we propose an innovative and contributive solution for the evaluation of the satellite-to-ground transmission efficiency and the scheme design of space-borne systems. The main novelty and contributions of this paper are summarized as follows:(1)The TEF of LEO EO satellites is proposed for CCM, VCM and ACM, which is defined by considering the link availability, useful data rate and the transmission arc segment of G/S. TEF could be utilized as a quantitative index for scheme selection.(2)A new analysis method is developed for the evaluation of the satellite-to-ground transmission efficiency, taking weather characteristics, transmission systems and other factors into account. Three G/Ss with representative weather conditions are selected in this paper in order to reach the conclusions for various application scenarios.

The rest of this paper is organized as follows: Section 2 presents the system model and configuration of LEO data transmission system, including the definition of TEF, the optimum selection of MCSs and other simulation parameters. In Section 3, the satellite-to-ground data transmission efficiency of CCM, VCM and ACM are analyzed quantitatively. Section 4 compares the transmission efficiency and completes the scheme design of satellite-to-ground data transmission. Section 5 concludes the paper.

## 2. System Model and Configuration

### 2.1. Definition of TEF

The satellite-to-ground data transmission of LEO EO satellites is intermittent. In the limited transmission arc segment, the high-rate data transmission is required to dump high-volume of payload data. If only pursuing high link availability, sufficient margins need to be reserved to compensate the atmospheric attenuation especially for small elevation angles, which will result in lower useful data rate and could not meet the requirement of high-rate data transmission.

For a G/S of LEO EO satellites, the transmission capacity of satellite-to-ground link is measured by the total data amount that can be received correctly in the transmission arc segment of a repeat period. However, different locations of G/S will lead to different total durations of transmission arc segments.

The application characteristics of CCM, VCM and ACM are different:(1)The MCS and useful data rate of CCM are fixed in all transmission arc segments.(2)VCM can complete MCS switching at some fixed elevation angles threshold, increase the MCS and change the useful data rate with the increase of elevation angles.(3)ACM can dynamically adjust MCS according to the state of satellite-to-ground link. Even at a specific elevation, any MCS may be used. It can be considered to use the maximum MCS first, whose link availability is minimum but the useful data rate is maximum, and then select a smaller MCS in different link availability intervals with the increase of link availability.

The TEF could be defined as “the product of link availability and average useful data rate”, which clarifies the weighted average effect of link availability on the useful data rate. Beyond the time corresponding to link availability (i.e., outage period), the bit error rate (BER) does not meet the requirement and the desired remote sensing data cannot be recovered correctly.

Therefore, the CCM TEF of LEO EO satellite for a single G/S is defined as:(1)$$E_{\mathrm{CCM}}=A_{L}\cdot R_{b}$$
where $A_{L}$ is the link availability and $R_{b}$ is the useful data rate. Both parameters are related to the MCS.

Then, the VCM TEF of LEO EO satellite for a single G/S is modified and defined as:(2)$$E_{\mathrm{VCM}}=A_{L}\cdot \frac{\sum_{i=1}^{N}\left[\int_{t_{\mathrm{start}}(i)}^{t_{\mathrm{end}}(i)}R_{b}(t)dt\right]}{\sum_{i=1}^{N}\left[t_{\mathrm{end}}(i)-t_{\mathrm{start}}(i)\right]}$$
where $A_{L}$ is the link availability (related to the selected initial minimum MCS), $R_{b}(t)$ is the useful data rate (Mbps) at time *t*, $t_{\mathrm{start}}(i)$ and $t_{\mathrm{end}}(i)$ are the start time and the end time of the i-th transmission arc segment, respectively, and *N* is the number of data transmission arc segments in a repeat period.

Furthermore, the ACM TEF of LEO EO satellite for a single G/S is modified and defined as:(3)$$E_{\mathrm{ACM}}=\frac{\sum_{i=1}^{N}\left\{\int_{t_{\mathrm{start}}(i)}^{t_{\mathrm{end}}(i)}\left[A_{L}(8)\cdot R_{b}(8,t)+\sum_{k=7}^{9-K}\left(A_{L}(k)-A_{L}(k+1)\right)\cdot R_{b}(k,t)\right]dt\right\}}{\sum_{i=1}^{N}\left[t_{\mathrm{end}}(i)-t_{\mathrm{start}}(i)\right]}$$
where $R_{b}(k,t)$ is the useful data rate of the *k*-th MCS at time t (*k* = 1 for MCS 1, 2 for MCS 3, 3 for MCS 6,…, 8 for MCS 28 in this paper), K is the number of MCSs used in ACM system and $A_{L}(k)$ is the link availability of the *k*-th MCS.

The integral of Equations (2) and (3) needs to be discretized and changed to cumulative summation through equal interval sampling, that is:(4)$$E_{\mathrm{VCM}}=A_{L}\cdot \frac{\sum_{i=1}^{N}\left[\sum_{j=1}^{M_{i}}R_{b}(t_{ij})\Delta T\right]}{\sum_{i=1}^{N}\left[t_{\mathrm{end}}(i)-t_{\mathrm{start}}(i)\right]}$$
(5)$$E_{\mathrm{ACM}}=\frac{\sum_{i=1}^{N}\left\{\sum_{j=1}^{M_{i}}\left[A_{L}(8)\cdot R_{b}(8,t_{ij})+\sum_{k=7}^{9-K}\left(A_{L}(k)-A_{L}(k+1)\right)\cdot R_{b}(k,t_{ij})\right]\Delta T\right\}}{\sum_{i=1}^{N}\left[t_{\mathrm{end}}(i)-t_{\mathrm{start}}(i)\right]}$$
where ΔT is the sampling interval, $t_{ij}$ is the j-th sampling time of the i-th transmission arc segment, $M_{i}$ is the number of sampling points of the i-th transmission arc segment and $R_{b}(k,t_{ij})$ is the useful data rate of the *k*-th MCS at the j-th sampling time of the i-th data transmission arc segment.

### 2.2. Modulation and Coding Mode Selection

Higher-order modulation and higher-rate channel coding can increase data transmission efficiency when the channel has sufficient SNR condition. In order to provide flexibility and fully use of different channel scenarios, DVB-S2 supports 28 kinds of MCSs with the span of demodulation threshold up to 21 dB. In order to avoid the frequent switching of MCSs and reduce the complexity of system implementation and simulation analysis appropriately, eight kinds of MCSs are selected in this paper, as shown in Table 1 [24].

### 2.3. Link Budget Parameters

This work adopts one sun-synchronous circular EO satellite with orbital height of 500 km [1], whose repeat period is 31 days. For all scenarios investigated herein, three typical G/Ss have been selected in China: Kashgar (latitude 39.50° N, longitude 75.92° E), Beijing (latitude 40.45° N, longitude 116.86° E) and Sanya (latitude 18.31° N, longitude 109.30° E), which are representative of temperate continental, temperate monsoon and tropical monsoon climate conditions, respectively [25]. For Kashgar, Beijing and Sanya G/S, the annual probability of rain is 0.38%, 1.94% and 2.46%, and the point rainfall rate for 0.01% of an average year is 9.42 mm/h, 42.56 mm/h and 81.09 mm/h, respectively. Therefore, the selected G/Ss can be representative of three typical states of drought, moderate rainfall and abundant rainfall in China.

Based on engineering experience and on a technical basis [26,27,28,29,30], the link budget parameters are given in Table 2. The satellite-to-ground data transmission link is assumed as an additive white Gaussian noise (AWGN) channel in this paper.

### 2.4. Characteristic Analysis of Transmission Arc Segment

The ground track of sun-synchronous circular orbit has repetition characteristics, and the repeat period is 31 days when the altitude is 500 km. Figure 1 illustrates the LEO EO satellite passing over the three G/Ss in 31 days, where the red, blue and yellow orbits correspond to pass over the Kashgar G/S, Beijing G/S and Sanya G/S, respectively. Considering the amount of simulation calculation and the attenuation characteristics of dynamic channel model, the sampling interval during transmission is set to 1 s. The quantitative analysis results are shown in Table 3.

It can be seen from Table 3 that:(a)The statistical transmission arc segment characteristics of the aforementioned three G/Ss (minimum duration, maximum duration and mean duration) are relatively close and have little relationship with G/S locations.(b)Kashgar G/S and Beijing G/S with similar latitude have similar transmission arc segment characteristics.(c)Since Sanya G/S has lower latitude, there are relatively fewer opportunities for LEO satellites to establish data transmission link with it. The number and the total duration of transmission arc segments are less than Kashgar G/S and Beijing G/S, with a relative proportion of about 80%.

By calculating the duration falling into the inter cell range within elevation angles from 5° to 90° at an interval of 1°, the results are clearly shown in Figure 2. It can be seen that the distribution trend of three G/Ss is similar, which decrease rapidly with the increase of elevation angles, and a large number of transmission arc segments are located in the area with small elevation angles.

Figure 3 shows the cumulative distribution of transmission arc segments from the minimum 5° elevation angle. It can be seen that the data laws of three G/Ss are basically the same, and the proportions below 10°, 20° and 30° are about 36%, 72% and 86%, respectively. In order to improve the transmission efficiency, we should make full use of the long transmission arc segments with small elevation angle and increase the useful data rate.

### 2.5. Allowable Atmospheric Attenuation Value of G/S

Ignoring the atmospheric attenuation, for the eight MCSs selected in Section 2.2, the maximum *E*_S_/*N*_0_ available at the G/S and the demodulation threshold values considering total degradation are shown in Figure 4. It can be seen that as the elevation angle increases from 5° to 90°, the maximum *E*_S_/*N*_0_ in clear sky available at the G/S increase from 36.01 dB to 48.39 dB, which are beyond the threshold values of all MCSs. The variation range of *E*_S_/*N*_0_ in clear sky reaches 12.38 dB. Therefore, under this ideal condition, the reliable data transmission can be achieved through the satellite-to-ground link.

In the actual satellite-to-ground data transmission link from LEO EO satellites to G/S, the atmospheric attenuation is inevitable, and is closely related to the G/S location and the link availability. If only pursuing high link availability, sufficient margins need to be reserved to compensate the atmospheric attenuation. This will result in a waste of resource for LEO EO satellites, which do not need to transmit at any time. In practical engineering applications, it is necessary to make tradeoff between implementation cost and link availability.

From Figure 4, the maximum allowable atmospheric attenuation values of the eight selected MCSs at different elevation angles can be obtained, as shown in Figure 5, which can be used for link availability analysis.

### 2.6. Link Availability Analysis

The atmospheric attenuation represents the combined effect of rain, gas, clouds and scintillation, and it is a function of the location, elevation angle and link availability of G/S [31,32,33]. This paper calculates the atmospheric attenuation according to the latest updates of P Series radiowave propagation recommendations issued by the International Telecommunication Union (ITU). Rain attenuation and scintillation attenuation are calculated according to ITU-R P.618-13 [31]. Cloud attenuation is calculated according to ITU-R P.840-8 [32]. Gaseous attenuation is calculated according to ITU-R P.676-12 [33].

For a specific G/S of LEO EO satellites, the link condition is worst at the minimum 5° elevation angle, whose link availability directly determines the reliable data transmission capacity of the full transmission arc segment. When the location and elevation of G/S are determined, the link availability corresponding to the maximum allowable atmospheric attenuation value can be calculated according to the aforementioned three ITU recommendations. The link availability, link unavailability and outage period of Kashgar G/S, Beijing G/S and Sanya G/S are shown in Table 4. For a specific MCS and G/S, the summation of the link availability and the link unavailability is 100%, and the outage period is the product of link unavailability and 365 days (1 year).

As clearly shown in Table 4:(1)For a specific G/S, with the increase of MCS, the link availability decreases, but the link unavailability and the outage period increase because of the smaller allowable atmospheric attenuation for the larger MCS.(2)For a specific MCS, since Kashgar G/S is in the driest location while Sanya G/S is in the wettest location, Kashgar G/S provides the most link availability, the least link unavailability and outage period. However, the link availability of Sanya G/S decreases significantly, and the link unavailability and the outage period of Sanya G/S increase significantly.(3)For different G/S, with the increase of MCS, the absolute values of link availability decreased have apparent difference. Kashgar G/S with little rain changes slightly (only 0.165%), and Beijing G/S with moderate rainfall changes moderately (3.689%), but Sanya G/S with abundant rainfall changes greatly (19.392%).

## 3. Performance Simulation and Analysis

### 3.1. CCM Transmission Efficiency

Combined with the data in Table 1 and Table 4, the CCM TEF of Kashgar G/S, Beijing G/S and Sanya G/S can be obtained, as shown in Table 5. It can be seen that:(1)For a specific G/S, the CCM TEF increases with the increase of MCS, mainly due to the larger system spectral efficiency for larger MCS. However, compared with the lower MCS, the outage period is longer for higher MCS.(2)For a specific MCS, the CCM TEF decreases in the order of Kashgar G/S, Beijing G/S and Sanya G/S, mainly due to the difference of rainfall.

### 3.2. VCM Transmission Efficiency

The VCM system simplifies the atmospheric attenuation at different elevation angles to a fixed value [1] and needs to ensure that the link is available at the minimum 5° elevation angle. Only the change of free space loss caused by the change of distance between satellite and G/S is considered, and it is used as the basis for MCS switching.

For different initial minimum MCS of VCM system, the theoretical range of elevation angle corresponding to different set of MCSs can be calculated from Figure 5. In Figure 5, starting from the upper seven maximum allowable atmospheric attenuation values with 5° elevation angle, seven straight lines parallel to the horizontal axis are drawn and the intersections with the existing curves which correspond to the switching elevation angles of adjacent MCS are found out. For example, when the initial minimum MCS is 1, the switching elevation angles of adjacent MCS are shown in Figure 6. The results are independent of G/S locations and shown in Table 6.

The simulation results of VCM TEF of Kashgar G/S, Beijing G/S and Sanya G/S with different initial minimum MCS are shown in Table 5. It can be seen that:(1)For a specific G/S, the VCM TEF increases with the increase of initial minimum MCS, mainly because of the larger system spectral efficiency for larger MCS. However, compared with the lower initial minimum MCS, the outage period is longer for higher initial minimum MCS.(2)For a specific initial minimum MCS, the VCM TEF decreases in the order of Kashgar G/S, Beijing G/S and Sanya G/S, mainly due to the differences of rainfall.

### 3.3. ACM Transmission Efficiency

ACM system considers the comprehensive effect of simultaneous reducing of free space loss and atmospheric attenuation when the elevation angle increases. Combined with the data in Table 1, when the elevation angle is 5°, the theoretical range of link availability interval corresponding to different MCS in ACM system is shown in Table 7. The maximum MCS 28 is always used, and the link availability is determined by the selected minimum MCS. With the increase of the used minimum MCS, statistically, the link availability decreases gradually for a specific G/S, but the changes of each G/S are different.

When performing ACM transmission efficiency analysis, the largest MCS 28 is preferred (the link availability is the lowest, but the useful data rate is the highest), and then the MCS is gradually reduced until the selected minimum MCS. When the transmission rate is weighted average, the MCS 28 is weighted by its corresponding link availability, while other MCSs are weighted by the gaps of adjacent link availability.

As the elevation angle increases, the atmospheric attenuation decreases and the link availability of each MCS increases. The reliable data transmission capacity is assessed for the full transmission arc segment, so the link availability for CCM is determined only at 5° elevation, and it is not the weighted average for different elevation angles, which will be larger. If we recomputed the link budget with the larger weighted average link availability, the atmospheric attenuation at 5° elevation angle will be larger correspondingly, which would result in dissatisfied inserted system margin (less than 3 dB or even negative), and this is unacceptable.

ACM is similar with CCM. In order to unify the comparison standard for CCM, VCM and ACM, the maximum link availability of ACM system is fixed as the link availability of the selected minimum MCS at 5° elevation angle. In this way, with the increase of elevation angle, the link availability of larger MCS is increasing. Starting from the selected minimum MCS, it is gradually not used, and finally only the largest MCS 28 needs to be selected.

Taking the minimum MCS 1 as an example with the parameters in Table 2, the link availabilities of each MCS at different elevation angles are illustrated in Figure 7. The relationship between link availability and elevation angle is derived from the link budget computation widely used in satellite-to-ground data transmission systems.

For different minimum MCS, the simulation results of ACM TEF of Kashgar G/S, Beijing G/S and Sanya G/S are shown in Table 5. It can be seen that:(1)For a specific G/S, the ACM TEF decreases with the increase of minimum MCS. This is due to the fact that the number of available MCS is reduced, and without the lower MCS, it may not be possible to cope with the larger atmospheric attenuation.(2)For a specific minimum MCS, the ACM TEF decreases in the order of Kashgar G/S, Beijing G/S and Sanya G/S, mainly due to the differences of rainfall.

## 4. Comparison and Scheme Design

### 4.1. Comparison and Analysis of Satellite-to-Ground Transmission Efficiency

According to the analysis in Section 3, the ACM, VCM and CCM TEF of each G/S are illustrated on one figure for comparison, as shown in Figure 8. For any G/S, it can be seen that:(1)For a specific MCS (initial minimum MCS for VCM and minimum MCS for ACM), the TEF increases in the order of CCM, VCM and ACM, because of the possibility to vary the MCS according to the available *E*_S_/*N*_0_ with VCM and ACM.(2)The TEF increases with the increase of CCM’s MCS or VCM’s initial minimum MCS. As shown in Table 1; Table 4, for CCM, the useful data rate increases rapidly with the increase of MCS, but the link availability decreases relatively slowly. So, the CCM TEF, which is the product of the link availability and the useful data rate, still increases. VCM is similar with CCM.(3)The TEF decreases with the increase of ACM’s minimum MCS. This is because the available number of MCSs decreases when the minimum MCS increases, and it is not always possible to find a suitable MCS to cope with the bigger atmospheric attenuation. For ACM, when encountering bad weather, such as rainstorms, the atmospheric attenuation is big, and the receiving *E*_S_/*N*_0_ at the G/S is small. If the smallest MCS is dropped, the bit error rate for other MCSs will be large. Additionally, the remote sensing data cannot be recovered correctly. So, if the smaller MCSs in ACM system were used, the probability of recovering data correctly would become larger, and the ACM TEF should also be larger.

It is important to note that the maximum transmission efficiency is obtained for ACM used at its best, which is with all the MCSs. So, if ACM is adopted, all the MCSs should be chosen, where the smallest MCS is used to cope with the larger atmospheric attenuation when encountering bad weather, such as a rainstorm, while the largest MCS is used to obtain a larger useful data rate and transmit as much data as possible during clear sky.

To analyze the transmission efficiency improvement quantitatively, the TEF of VCM, ACM and CCM are compared, respectively. The improvement percentages are obtained, as shown in Table 8 and Figure 9. For any G/S, it can be seen that:(1)Compared with CCM, VCM can obtain a maximum efficiency improvement of about 70%. The efficiency improvement proportion has little to do with the G/S position and rainfall, and gradually decreases with the increase of MCS.(2)Compared with CCM, ACM can obtain a maximum efficiency improvement of more than 796%, and the efficiency improvement proportion gradually decreases with the increase of MCS.(3)Compared with VCM, ACM can obtain a maximum efficiency improvement of more than 426%, and the efficiency improvement proportion gradually decreases with the increase of MCS.

In Ref. [21], for Svalbard G/S, whose annual probability of rain is 1.44%, the average data rate in ACM system is 1490 Mbps if the minimum link availability is 99.5%. However, if the considering approach in this paper is adopted, for Beijing and Sanya G/S, whose annual probability of rain is 1.94% and 2.46%, respectively (all bigger than the one of Svalbard G/S), the maximum average data rate in ACM system is 2644.63 Mbps and 2468.56 Mbps, respectively, and they are all much bigger.

### 4.2. Scheme Design of Satellite-to-Ground Data Transmission

In engineering applications, it is necessary not only to improve the average useful data rate, but also to minimize the outage period (improve link availability) to avoid excessive impact on user data receiving tasks.

As mentioned above, the maximum CCM TEF of the aforementioned three G/Ss are all achieved when the MCS is 28. For a specific G/S, by comparing the TEF of VCM and ACM with the one of CCM of MCS 28, the data-transmission scheme (CCM, VCM or ACM) could be decided. The comparison results are given in Table 9.

Compared with the CCM of MCS 28, if VCM is adopted:(1)The VCM TEF of Kashgar G/S and Beijing G/S are even smaller. However, the link availability is only increased from 99.820% to 99.881% for Kashgar G/S and from 95.527% to 96.452% for Beijing G/S. According to the principle of “maximizing transmission efficiency”, VCM is not recommended.(2)The VCM TEF of Sanya G/S is slightly larger (only increased by 0.74%) and the link availability is increased from 74.207% to 78.839%. The improvement is limited. Therefore, VCM is also not recommended.

Compared with the CCM of MCS 28, if the ACM of minimum MCS 1 is adopted:(1)For Kashgar G/S with little rain, the link unavailability is reduced from 0.180% to 0.015%, the average annual outage period is reduced from 0.66 days to 0.05 days, which can be almost ignored, and the TEF is only increased by 0.16%. Therefore, according to the principle of “simplification of satellite ground interaction”, the CCM of MCS 28 is more suitable for use.(2)For Beijing G/S with moderate rainfall, the link unavailability is reduced from 4.473% to 0.784%, and the average annual outage period is reduced from 16.33 days to 2.86 days. The reduction proportion is as high as 82.47%, and the TEF is increased by 3.62%, making ACM more suitable for use.(3)For Sanya G/S with abundant rainfall, the link unavailability is reduced from 25.793% to 6.401% and the average annual outage period is reduced from 94.14 days to 23.36 days. The reduction proportion is as high as 75.18%, and the TEF is increased by 24.51%, making ACM more suitable for use.

## 5. Conclusions

In this paper, the ACM based on the DVB-S2 standard is developed and utilized to improve the data transmission efficiency of LEO EO satellites. Additionally, the ACM could adaptively adjust the MCS and make better use of link resources under different weather conditions and elevation angles. The TEF is proposed to realize the scheme design and applications of Ka-band data transmission systems for LEO satellites. The simulation results show that Kashgar G/S, with little rain and representative of temperate continental climate conditions, is more suitable for CCM of MCS 28. Beijing G/S, with moderate rainfall and representative of temperate monsoon climate conditions, and Sanya G/S, with abundant rainfall and representative of tropical monsoon climate conditions, are more suitable for ACM of minimum MCS 1.

The proposed design and analysis method in this paper could comprehensively evaluate the satellite-to-ground transmission efficiency. It is calculated according to the change of free space loss and atmospheric attenuation with the elevation angle distribution of transmission arc segment in a repeat period, which is in accordance with engineering practice. This model and method could find important applications in the system design and implementation of Ka-band ACM satellite-to-ground data transmission links.

## Figures and Tables

**Figure 1 sensors-22-05423-f001:**
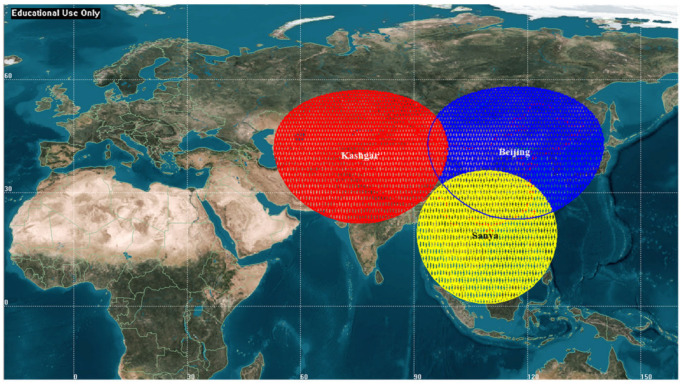
Schematic diagram of transmission arc segment in 31 days.

**Figure 2 sensors-22-05423-f002:**
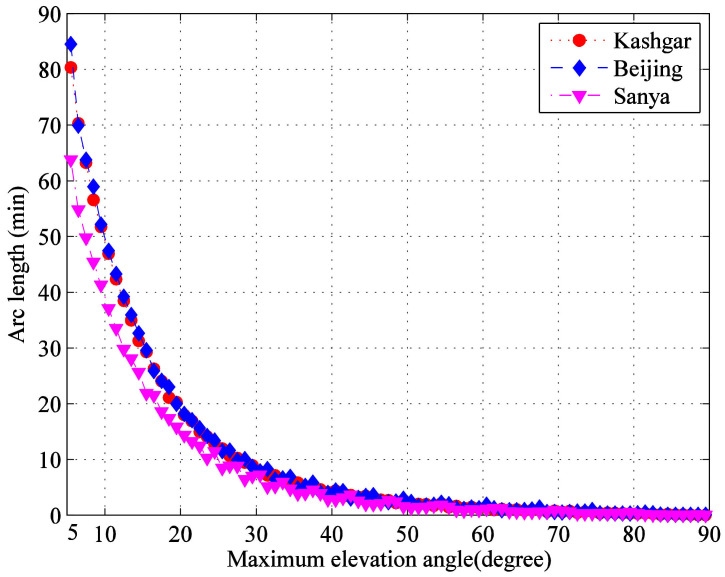
Transmission arc segment duration distributions of every 1° elevation angle.

**Figure 3 sensors-22-05423-f003:**
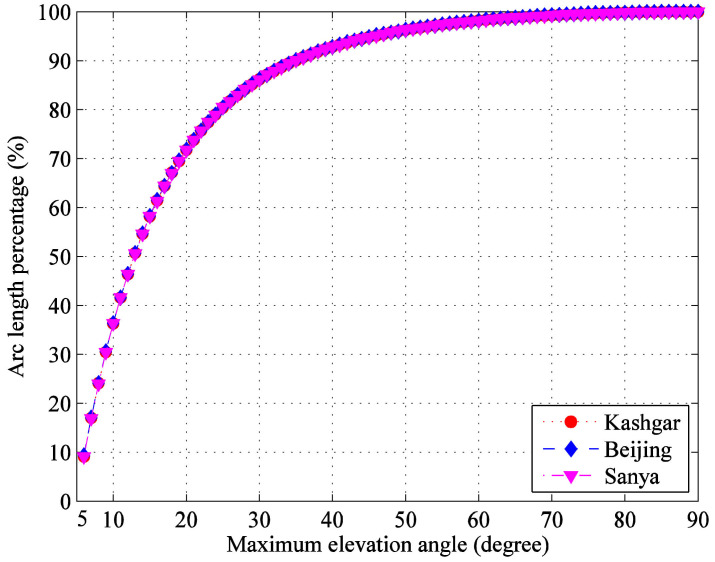
Cumulative distribution of transmission arc segments from the minimum 5° elevation angle.

**Figure 4 sensors-22-05423-f004:**
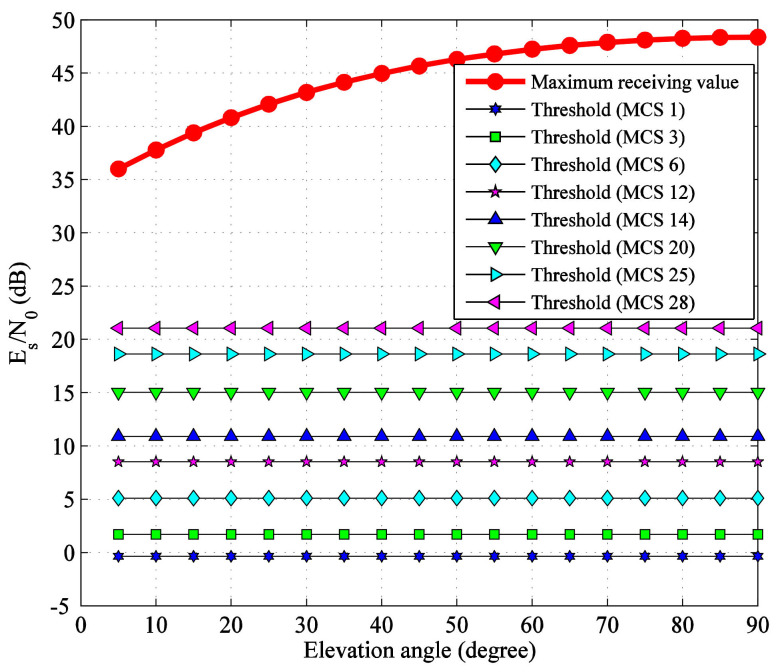
Comparison between receiving value in clear sky and threshold of *E*_S_/*N*_0_.

**Figure 5 sensors-22-05423-f005:**
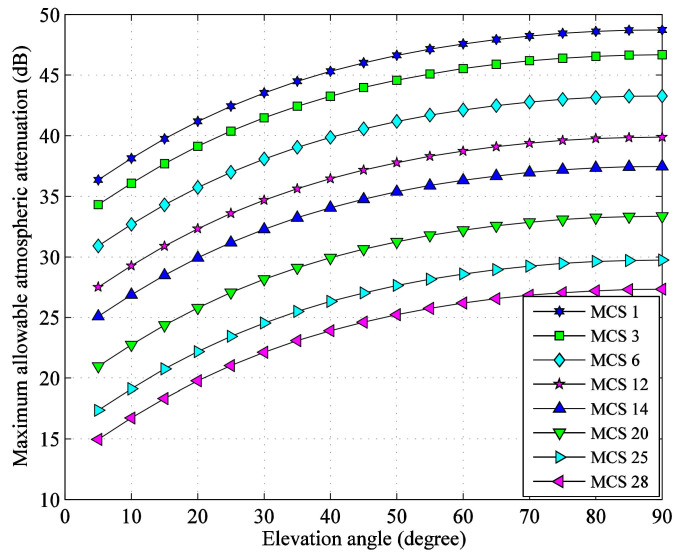
Maximum allowable atmospheric attenuation.

**Figure 6 sensors-22-05423-f006:**
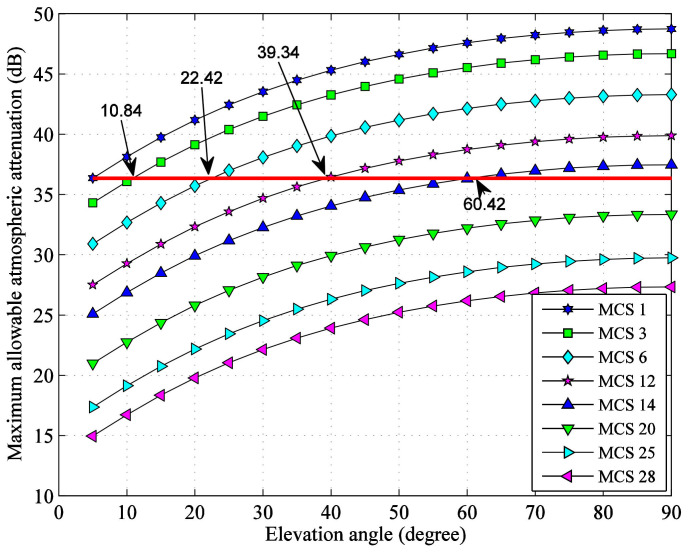
Switching elevation angle of adjacent MCS (initial minimum MCS 1).

**Figure 7 sensors-22-05423-f007:**
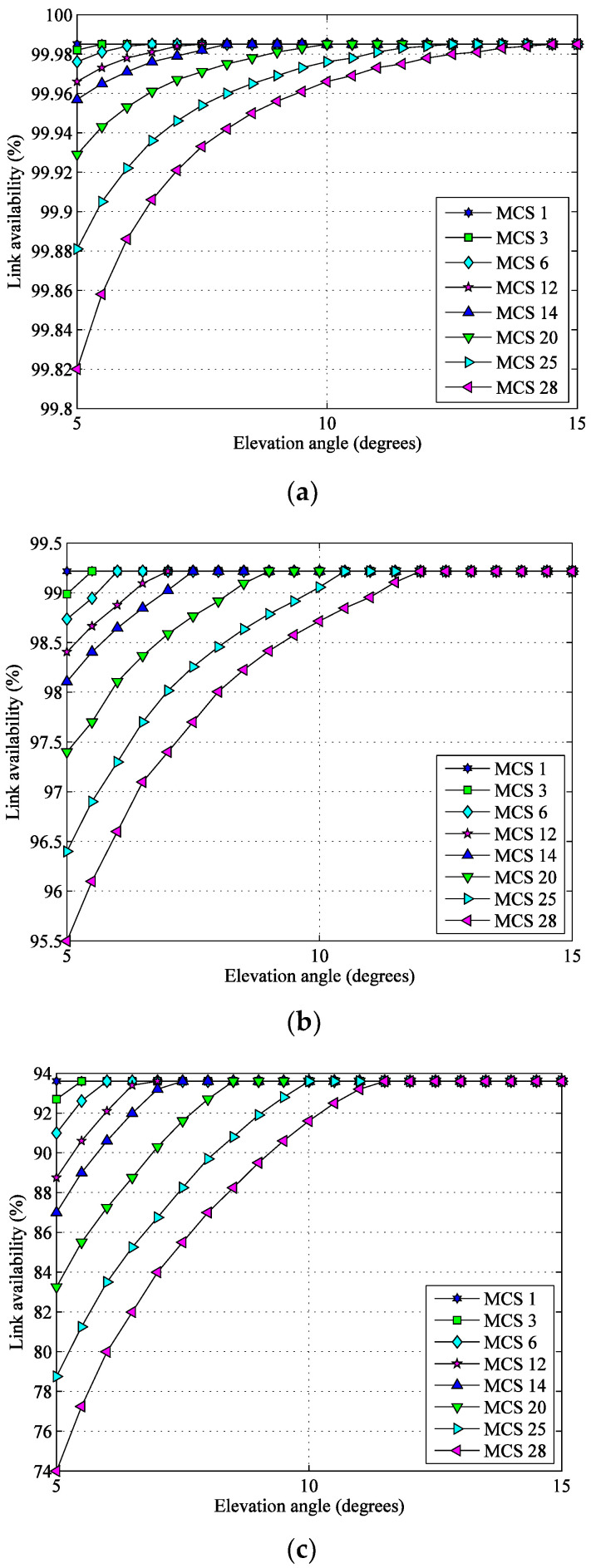
Link availability of each MCS at different elevation angles. (**a**) Kashgar G/S; (**b**) Beijing G/S; (**c**) Sanya G/S.

**Figure 8 sensors-22-05423-f008:**
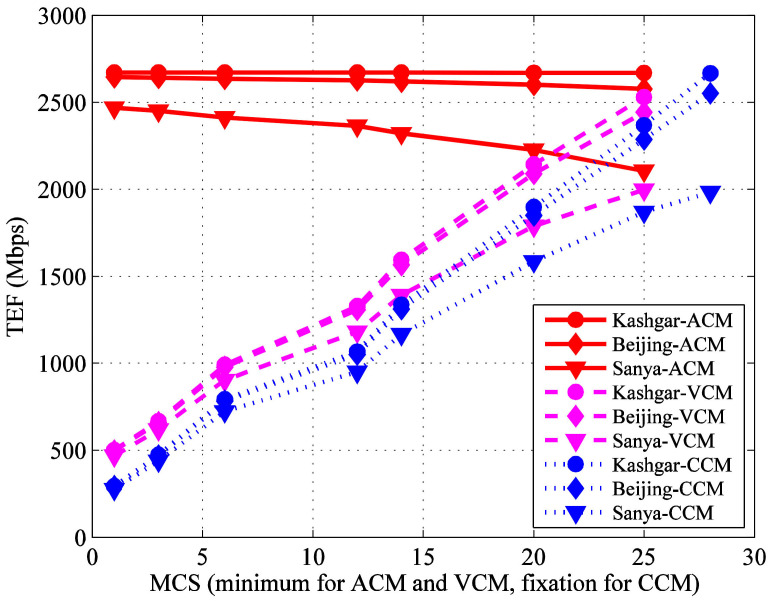
TEF comparison of different G/S adopting different transmission schemes.

**Figure 9 sensors-22-05423-f009:**
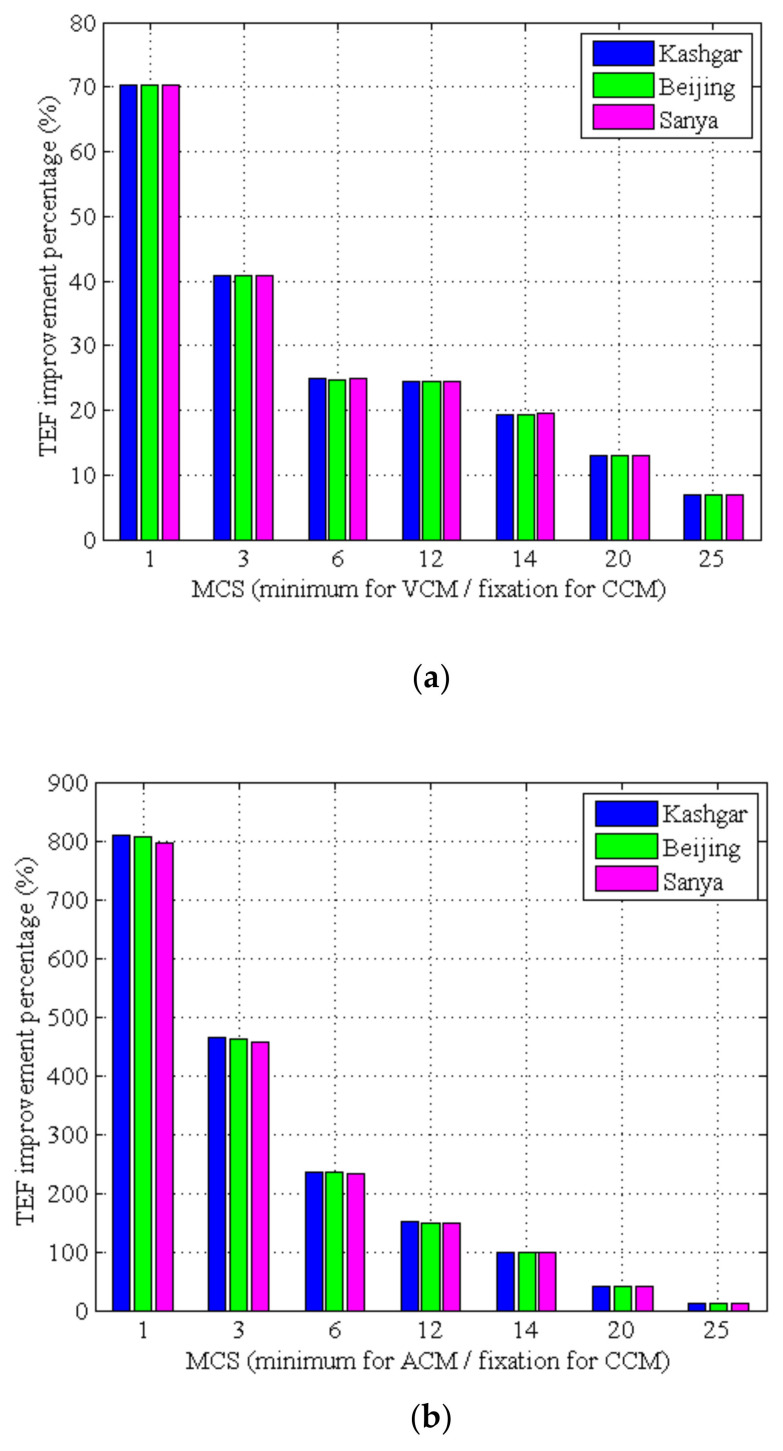
TEF improvement percentage. (**a**) VCM vs. CCM; (**b**) ACM vs. CCM; (**c**) ACM vs. VCM.

**Table 1 sensors-22-05423-t001:** Characteristic of MCSs recommended by DVB-S2.

MCS	Mode Name	Spectral Efficiency (bit/s/Hz)	*E*_S_/*N*_0_ Threshold (dB)	Useful Data Rate (Mbps)
1	QPSK 1/4	0.490 243	−0.35	294.15
3	QPSK 2/5	0.789 412	1.70	473.65
6	QPSK 2/3	1.322 253	5.10	793.35
12	8PSK 3/5	1.779 991	8.50	1067.99
14	8PSK 3/4	2.228 124	10.91	1336.87
20	16APSK 4/5	3.165 623	15.03	1899.37
25	32APSK 4/5	3.951 571	18.64	2370.94
28	32APSK 9/10	4.453 027	21.05	2671.82

**Table 2 sensors-22-05423-t002:** System parameters and link budget characteristics of the end-to-end simulation.

Parameter	Value
Orbit type	LEO sun-synchronous circular orbit
Satellite altitude	500 km
G/S location	Kashgar, Beijing, Sanya
Downlink carrier frequency	26.64 GHz
Polarization	Circular
Roll-off factor	0.2
Symbol rate	600 Mbaud
Maximum EIRP	47 dBW
G/S pointing error loss	1 dB
Polarization mismatch loss	0.5 dB
G/S G/T	40 dB/K
Inserted system margin	3 dB

**Table 3 sensors-22-05423-t003:** Statistic of satellite-to-ground data transmission arc segment in 31 days.

G/S	Number	Duration/min			
		Minimum	Maximum	Mean	Total
Kashgar	123	0.99	9.12	7.19	883.86
Beijing	126	1.37	9.13	7.16	901.75
Sanya	98	1.13	9.05	7.14	700.19

**Table 4 sensors-22-05423-t004:** Link availability, link unavailability and outage period of satellite-to-ground data transmission link at 5° elevation angle.

MCS	Link Availability/%			Link Unavailability/%			Outage Period/d		
	Kashgar	Beijing	Sanya	Kashgar	Beijing	Sanya	Kashgar	Beijing	Sanya
1	99.985	99.216	93.599	0.015	0.784	6.401	0.05	2.86	23.36
3	99.982	98.986	92.751	0.018	1.014	7.249	0.07	3.70	26.46
6	99.976	98.735	91.034	0.024	1.265	8.966	0.09	4.62	32.73
12	99.966	98.406	88.928	0.034	1.594	11.072	0.12	5.82	40.41
14	99.957	98.108	87.179	0.043	1.892	12.821	0.16	6.91	46.80
20	99.929	97.405	83.412	0.071	2.595	16.588	0.26	9.47	60.55
25	99.881	96.452	78.839	0.119	3.548	21.161	0.43	12.95	77.24
28	99.820	95.527	74.207	0.180	4.473	25.793	0.66	16.33	94.14

**Table 5 sensors-22-05423-t005:** CCM, VCM and ACM TEF (Mbps).

MCS	CCM			VCM			ACM		
	Kashgar	Beijing	Sanya	Kashgar	Beijing	Sanya	Kashgar	Beijing	Sanya
1	294.11	291.84	275.32	501.12	497.04	469.05	2671.21	2644.63	2468.56
3	473.56	468.85	439.32	666.96	659.87	618.64	2671.15	2639.80	2450.48
6	793.16	783.31	722.22	989.98	977.43	901.88	2671.02	2634.28	2412.24
12	1067.63	1050.97	949.74	1327.71	1306.82	1181.04	2670.80	2626.65	2364.68
14	1336.30	1311.58	1165.47	1595.23	1565.33	1392.04	2670.58	2619.44	2320.68
20	1898.02	1850.08	1584.30	2143.19	2089.15	1789.49	2669.88	2601.79	2225.59
25	2368.12	2286.82	1869.23	2530.34	2443.33	1997.30	2668.63	2576.91	2105.92
28	2667.01	2552.31	1982.68	-	-	-	-	-	-

Note: The leftmost column is the initial minimum MCS for VCM and minimum MCS for ACM.

**Table 6 sensors-22-05423-t006:** Theoretical range of elevation angle corresponding to different MCS in VCM system (°).

MCS	Initial Minimum MCS							
	1	3	6	12	14	20	25	28
1	[5, 10.84)	-	-	-	-	-	-	-
3	[10.84, 22.42)	[5, 15.06)	-	-	-	-	-	-
6	[22.42, 39.34)	[15.06, 28.21)	[5, 15.06)	-	-	-	-	-
12	[39.34, 60.42)	[28.21, 41.73)	[15.06, 23.88)	[5, 11.93)	-	-	-	-
14	[60.42, 90]	[41.73, 90]	[23.88, 47.1)	[11.93, 26.97)	[5, 17.49)	-	-	-
20	-	-	[47.1, 90]	[26.97, 48.85)	[17.49, 32.82)	[5, 15.75)	-	-
25	-	-	-	[48.85, 90]	[32.82, 48.85)	[15.75, 24.75)	[5, 11.93)	-
28	-	-	-	-	[48.85, 90]	[24.75, 90]	[11.93, 90]	[5, 90]

**Table 7 sensors-22-05423-t007:** Theoretical range of link availability and weighted probability of each MCS in ACM system at 5° elevation angle (%).

MCS	Range of Link Availability			Weighted Probability		
	Kashgar	Beijing	Sanya	Kashgar	Beijing	Sanya
1	(99.982, 99.985]	(98.986, 99.216]	(92.751, 93.599]	0.003	0.230	0.848
3	(99.976, 99.982]	(98.735, 98.986]	(91.034, 92.751]	0.006	0.251	1.717
6	(99.966, 99.976]	(98.406, 98.735]	(88.928, 91.034]	0.010	0.329	2.106
12	(99.957, 99.966]	(98.108, 98.406]	(87.179, 88.928]	0.009	0.298	1.749
14	(99.929, 99.957]	(97.405, 98.108]	(83.412, 87.179]	0.028	0.703	3.767
20	(99.881, 99.929]	(96.452, 97.405]	(78.839, 83.412]	0.048	0.953	4.573
25	(99.82, 99.881]	(95.527, 96.452]	(74.207, 78.839]	0.061	0.925	4.632
28	[0, 99.82]	[0, 95.527]	[0, 74.207]	99.82	95.527	74.207

**Table 8 sensors-22-05423-t008:** TEF improvement percentage (%).

MCS	VCM vs. CCM			ACM vs. CCM			ACM vs. VCM		
	Kashgar	Beijing	Sanya	Kashgar	Beijing	Sanya	Kashgar	Beijing	Sanya
1	70.39	70.31	70.37	808.25	806.18	796.61	433.05	432.07	426.29
3	40.84	40.74	40.82	464.05	463.04	457.80	300.50	300.05	296.11
6	24.82	24.78	24.88	236.76	236.30	234.00	169.80	169.51	167.47
12	24.36	24.34	24.35	150.16	149.93	148.98	101.16	100.99	100.22
14	19.38	19.35	19.44	99.85	99.72	99.12	67.41	67.34	66.71
20	12.92	12.92	12.95	40.67	40.63	40.48	24.58	24.54	24.37
25	6.85	6.84	6.85	12.69	12.69	12.66	5.47	5.47	5.44

Note: The leftmost column is the initial minimum MCS for VCM and the minimum MCS for ACM.

**Table 9 sensors-22-05423-t009:** The TEF ratio of ACM or VCM relative to CCM of MCS 28 (%).

Minimum MCS	VCM			ACM		
	Kashgar	Beijing	Sanya	Kashgar	Beijing	Sanya
1	18.79	19.47	23.66	100.16	103.62	124.51
3	25.01	25.85	31.20	100.16	103.43	123.59
6	37.12	38.30	45.49	100.15	103.21	121.67
12	49.78	51.20	59.57	100.14	102.91	119.27
14	59.81	61.33	70.21	100.13	102.63	117.05
20	80.36	81.85	90.26	100.11	101.94	112.25
25	94.88	95.73	100.74	100.06	100.96	106.22

## Data Availability

The data that support the findings of this study are available from the corresponding author upon reasonable request.

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
