# Peer review of "A Transmission Efficiency Evaluation Method of Adaptive Coding Modulation for Ka-Band Data-Transmission of LEO EO Satellites"

_sensors, 2022, doi:10.3390/s22145423_

Round 1

Reviewer 1 Report

The topic of the paper and the result is interesting. However, there are some points to be clarified to be accepted as a publication. Please kindly consider the following suggestions.

  1. In the introduction, it is good that the authors already explained some previous work related to data-transmission performance. There is one sentence that highlighted the weakness of previous works which is “However, the correlation between transmission efficiency and link availability has not been noticed in the above papers.”. Please give more explanation why the correlation between transmission efficiency and link availability will have a significant impact on this study. 
  2. In sections 3.1 and 3.2,  the authors mention about outage period in the analysis of their results. However, there is no one result in table 3 showing about outage period result. Please give more explanation about it.
  3. For figure 1, there is no explanation of the parameter used in the simulation that produces figure 1. Please explain it.
  4. In Figure 2, it is interesting to see that for VCM and CCM, the value of TEF is increasing with the increase of MCS value. However, it is quite different for ACM. Please explain the interpretation and analysis of this result.
  5. The authors for most of the time just explained the interpretation of the results. Please give more analysis about the result. For example, why this could happen, what the consequence of the result is, etc. 

Reviewer 2 Report

Manuscript:
A Transmission Efficiency Evaluation Method of Adaptive Coding Modulation for Ka-band Data-transmission of LEO EO Satellites
submitted by:
Zhongguo Wang, Fan Lu, Dabao Wang, Xiao Zhang, Jionghui Li and Jindong Li

For low Earth orbit (LEO) Earth observation (EO) the orbit characteristics lead to limited time for satellite-to-ground data transmission. Technological advancements in satellite engineering have made it feasible to develop EO satellite systems with high spatial, spectral, temporal, and radiometric resolutions.

In the manuscript, the Authors propose an innovative and contributive solution for the evaluation of the satellite-to-ground transmission efficiency and the scheme design of space-borne systems. Their paper summarizes: the TEF of LEO EO satellites is proposed for CCM, VCM and ACM, which is defined by considering the link availability, useful data rate and transmission arc segment of G/S, three G/Ss with representative weather conditions are selected in order to reach the conclusions for various application scenarios anda new analysis method is developed for the evaluation of the satellite-to-ground transmission efficiency, taking weather characteristics, transmission system and other factors into account.  

The structure of the manuscript is considered and clear. In the introduction, the background and comprehensive review of the problem's literature were presented. The Authors present system model and configuration: definition of TEF, modulation and coding mode selection, link budget parameters. Performance simulation and analysis: CCM, VCM and ACM transmissions efficiency have been presented in tabular and graphic form. Conclusions, on the basis of the research.

1. The Authors submitted the mnuscript as an article, so, according to the instruction for authors, recommended length is more 16 pages. The article is too short.
2. Following suggestions should be taken into consideration (they are marked in the manuscript):
In equations (1)-(5) cross shold be repaced as a dot, cross is used for vector multiplication.
3. The manuscript seems to be converted from other article: references should be adjusted to instruction for Authors.
4. The manuscript should be reconsidered - extended.

Reviewer 3 Report

The topic of this paper is interesting and timely. The paper is well written and organized. The main reviewer concerns are summarized as follow,

1-) It is not clear what the main novelty of this work is. Point 1) and 2) are not contributions. Point 3) is a contribution but need to better explained and linked to the previous SA. It seems that only performance analysis of a well-known system is considered.

2) Improve the SoA by adding more recent references.

3) The considering approach is not compares with previous ones proposed in the literature.

4) Which channel model are you assumed?

Round 2

Reviewer 2 Report

A Transmission Efficiency Evaluation Method of Adaptive Coding Modulation for Ka-band Data-transmission of LEO EO Satellites

submitted by:

Zhongguo Wang, Fan Lu, Dabao Wang, Xiao Zhang, Jionghui Li and Jindong Li

Second review

I would like to thanks the Authors for taking into consideration my critical suggestions.

All detailed suggestions was taken into consideration and highlighted in the Response to Reviewre Comments. 

The manusctipt has been extended and meets the requirements for the Sensors Article.

I recommend the manuscript for publication in Sensors.

Reviewer 3 Report

The authors have satisfactory addressed most of the reviewer concerns.